# Soil Treatment with Nitric Oxide-Releasing Chitosan Nanoparticles Protects the Root System and Promotes the Growth of Soybean Plants under Copper Stress

**DOI:** 10.3390/plants11233245

**Published:** 2022-11-26

**Authors:** Diego G. Gomes, Tatiane V. Debiasi, Milena T. Pelegrino, Rodrigo M. Pereira, Gabrijel Ondrasek, Bruno L. Batista, Amedea B. Seabra, Halley C. Oliveira

**Affiliations:** 1Department of Agronomy, State University of Londrina (UEL), Celso Garcia Cid Road, Km 380, Londrina 86057-970, Brazil; 2Department of Animal and Plant Biology, State University of Londrina (UEL), Celso Garcia Cid Road, Km 380, Londrina 86057-970, Brazil; 3Center for Natural and Human Sciences, Federal University of ABC (UFABC), Avenida dos Estados, Saint Andrew 09210-580, Brazil; 4Department of Soil Amelioration, Faculty of Agriculture, University of Zagreb, 10000 Zagreb, Croatia

**Keywords:** S-nitrosothiol, metal stress, biopolymer, nanotechnology

## Abstract

The nanoencapsulation of nitric oxide (NO) donors is an attractive technique to protect these molecules from rapid degradation, expanding, and enabling their use in agriculture. Here, we evaluated the effect of the soil application of chitosan nanoparticles containing S-nitroso-MSA (a S-nitrosothiol) on the protection of soybeans (Glycine max cv. BRS 257) against copper (Cu) stress. Soybeans were grown in a greenhouse in soil supplemented with 164 and 244 mg kg^−1^ Cu and treated with a free or nanoencapsulated NO donor at 1 mM, as well as with nanoparticles without NO. There were also soybean plants treated with distilled water and maintained in soil without Cu addition (control), and with Cu addition (water). The exogenous application of the nanoencapsulated and free S-nitroso-MSA improved the growth and promoted the maintenance of the photosynthetic activity in Cu-stressed plants. However, only the nanoencapsulated S-nitroso-MSA increased the bioavailability of NO in the roots, providing a more significant induction of the antioxidant activity, the attenuation of oxidative damage, and a greater capacity to mitigate the root nutritional imbalance triggered by Cu stress. The results suggest that the nanoencapsulation of the NO donors enables a more efficient delivery of NO for the protection of soybean plants under Cu stress.

## 1. Introduction

Population growth over the last few decades has been expressive, creating a relationship between economic development, consumption of natural resources, and environmental degradation [1]. Among the possible means of degradation, the accumulation of metals, due to anthropogenic activities, stands out as one of the most harmful to human health [2]. In the agroecosystems, soil contamination through inappropriate practices, such as wastewater irrigation, the unsustainable use of pesticides and fertilizers, can induce the accumulation of metals [3] and their transfer to crops [4], creating a severe threat to the human food chain [5].

Among the essential metals, copper (Cu) has been used in agricultural practice for centuries, as an active ingredient of fungicides to enhance crop production by controlling plant diseases [6]. Inevitably, the usual, long-term foliar application of Cu-based fungicides and the restricted Cu mobility in the soil have promoted the Cu accumulation in the soil surface layers, as a result of the direct application, drift, or dripping of excess sprays from the leaves [7]. The accumulation of high Cu levels has been a frequently reported problem for vineyards and orchard soils [8]. For different plant species, including crops, the effects of Cu toxicity on the plant metabolism are commonly reported [9]. Previously, we demonstrated that increased soil Cu concentration delays the germination and negatively affects the initial development of soybean seedlings [10]. However, soybean plants develop some tolerance mechanisms (such as a low translocation of Cu to the leaves, a higher water use efficiency, and an elevated carboxylation efficiency) that contributes to the partial alleviation of Cu-induced stress [11].

Recently, the role of nitric oxide (NO) as a signaling molecule in the induction of plant tolerance to metal stress, has been confirmed. For instance, the treatment with NO donors (mainly sodium nitroprusside—SNP) has been shown to protect plants against oxidative damage induced by metal stress [12]. The protection occurs through the induction of the antioxidant response and the limitation of the amount of metal accumulated and translocated from the root to the shoot. [13]. S-nitrosothiols (RSNOs), formed by the S-nitrosation of thiol groups, are considered as suitable NO carriers and transporters for biological applications [14]. The treatment with NO donors has been used as an effective technique to increase the plant’s endogenous NO content and provoke the NO biological effects. However, such molecules are rapidly degraded and sensitive to environmental factors. They can release NO too quickly and/or generate toxic by-products, compromising the desired signaling action of NO on the target plants [15].

To reduce the limitations of the application of NO donors, chitosan polymeric nanoparticles have been developed as carrier systems for these molecules [15], aiming at protecting plants against different stresses, such as salinity [16,17]; photoinhibition [18], and drought [14,19,20]. Furthermore, NO-releasing chitosan nanoparticles have been suggested to be the most suitable candidates to alleviate the toxicity caused by metal in plants, owing to their unique properties, managing to release NO efficiently and with a prolonged duration [21].

Chitosan (CS) is a polymer derived from chitin and is considered a very versatile, biocompatible, biodegradable, and non-toxic biomaterial. It has mucoadhesive properties, facilitating the transport of bioactive compounds across the cell membranes, which implies a great potential for use in the agroindustry [22,23]. Moreover, CS can elicit plant defense responses against stress and has biostimulant characteristics [24]. Its application can induce the synthesis of different intracellular messengers, such as H_2_O_2_ and NO, acting on the stress signaling and triggering the protection responses [24]. Although the association between CS nanoparticles and the NO donor has shown the potential to confer protection against some abiotic stresses, protection against metal stress has not yet been tested. This is particularly relevant, as metal cations (including Cu^2+^) are known to accelerate RSNO decomposition, thus compromising the NO signaling effects [25].

The objective of the present work was to evaluate the protective effect of the soil treatment with CS/sodium tripolyphosphate nanoparticles containing the NO donor S-nitroso-mercaptosuccinic acid (S-nitroso-MSA) in the protection of soybean plants (Glycine max cv. BRS 257) from Cu stress, compared to the free form of S-nitroso-MSA.

## 2. Results

### 2.1. Characterization of the Nanoparticles and Encapsulation Efficiency of MSA

The MSA_NP showed a hydrodynamic size of 128.5 ± 10.1 nm, with a PDI of 0.29 ± 0.010 and a zeta potential of +18.2 ± 0.1 mV, as assayed by DLS. Moreover, Figure 1 shows the representative TEM image of MSA_NP at a solid state, which exhibits a spherical shape with an average size of 12.3 ± 0.9 nm and 77.4% of particles with a size lower than 12.8 nm. In addition, the encapsulation efficiency of MSA into the CS nanoparticles was found to be 93.5 ± 0.01%. The concentration of MSA incorporated into the CS nanoparticles (1 mM) is in the range of the nanoencapsulated NO donor concentrations previously used for plant applications [16,19]. 

### 2.2. Morphophysiological Analyses

Regarding the plant growth variables, no significant differences were found between the control and water treatments under moderate Cu stress (Table 1). Compared with the control and water, S-nitroso-MSA_NP stood out in almost all variables evaluated, except LA. S-nitroso-MSA_NP promoted also higher SL than S-nitroso-MSA and MSA_NP. For all other variables, no significant differences were found between S-nitroso-MSA_NP and S-nitroso-MSA. Under severe Cu stress, significant differences were observed between the control and water, with a decrease in all growth parameters (Table 1). All formulations (S-nitroso-MSA_NP, S-nitroso-MSA and MSA_NP) increased the parameters related to the root growth (RL and RDW), compared to the water treatment, but only S-nitroso-MSA_NP affected positively the SL and SDW of Cu-stressed plants. 

About the visual aspect of the root system, few differences were observed in plants grown under moderate Cu stress (Figure 2a), whereas the roots of plants grown under severe Cu stress showed more pronounced differences (Figure 2b). In particular, S-nitroso-MSA_NP was the only treatment to promote a root system closer to what was observed for the control (Figure 2b).

The PSII activity was affected differently by the treatments at each of the Cu concentrations in the soil (Figure 3). For F_v_/F_0_, no significant differences were observed between the treatments under moderate Cu stress (Figure 3a). Under severe Cu stress, it was possible to observe a 21% reduction in water, while the other treatments did not differ from the control (Figure 3b). The rETR was affected in both cultivation conditions. While the water treatment showed a reduction of 16%, the other treatments did not differ from the control under moderate Cu stress (Figure 3c). Under severe Cu stress, only S-nitroso-MSA showed a significant reduction (15%), compared to the control (Figure 3d).

Figure 4 presents the values that show significant alterations in the leaf gas exchange of treated plants. Compared to the control, the supplementation of the soil with both Cu concentrations caused a decrease in *A*, but this effect was prevented by the treatments with S-nitroso-MSA_NP and *S*-nitroso-MSA (Figure 4a,b). For *g*_s_, no significant differences were found among the treatments under moderate Cu stress (Figure 4c). Under severe Cu stress, only S-nitroso-MSA_NP caused no significant reduction in *g*_s_, compared to the control. On average, plants treated with S-nitroso-MSA and MSA_NP showed a decrease of 23% in the *g*_s_, compared to the control, while the inhibition was of 37% in the water treatment (Figure 4d). 

Regarding *k*, there is no significant difference between the treatments and the control under moderate Cu stress, in spite of the increase in *k* induced by S-nitroso-MSA_NP, compared to the water treatment (Figure 5a). Under severe Cu stress, the S-nitroso-MSA_NP and S-nitroso-MSA treatments completely prevented the Cu-induced decrease in *k*. On average, the water treatment and MSA_NP showed a 16% reduction in *k,* compared to the control (Figure 5b).

### 2.3. Biochemical Analyses

The plants treated with S-nitroso-MSA_NP and cultivated under moderate Cu stress showed an increase of 230% in the root RSNO content, compared to the other treatments (Figure 6a). Under severe Cu stress, the pattern observed was the same, as S-nitroso-MSA_NP was the treatment that provided the highest RSNO content in the roots (increase of 93%, compared to the other treatments) (Figure 6b).

Regarding the parameters that indicate the occurrence of oxidative stress, Figure 7a,c shows that the tested formulations did not affect the root H_2_O_2_ and CD levels, compared to the water treatment under moderate Cu stress. In contrast, under severe Cu stress, it was possible to verify an increase in H_2_O_2_ content in the water (107%) and MSA_NP (53%) treatments (compared to the control), which was prevented by S-nitroso-MSA_NP and S-nitroso-MSA (Figure 7b). For CD, S-nitroso-MSA_NP was the only treatment that prevented the Cu-induced increase in this marker of oxidative stress (Figure 7d).

When evaluating the antioxidant enzyme activity, S-nitroso-MSA tended to show the lowest SOD activity in both experiments (Figure 8a,b). For the POD activity, the treatments did not differ from the control under moderate Cu stress (Figure 8c). Severe Cu stress induced nearly doubled the POD activity in the water treatment compared to the control. This increment in the POD activity was prevented only by S-nitroso-MSA_NP (Figure 8d).

For the APX activity, the only effect detected was the decrease induced by MSA_NP, compared to the other treatments under moderate Cu stress (Figure 9a). In contrast, MSA_NP provided the highest CAT activity, with significant differences, compared to the control and S-nitroso-MSA treatments in the same experiment (Figure 9c). Under severe Cu stress, no significant differences were found among the treatments for the APX and CAT activities (Figure 9b,d).

### 2.4. Nutrient Analysis

It was possible to observe significant differences in the root Cu content between the treatments in both experiments (Figure 10). The moderate and severe Cu stress increased by nearly 100, and by 260% for the Cu content in the roots, compared to the control. Small differences in this parameter were observed among the formulations tested. S-nitroso-MSA_NP provided the highest Cu levels in the root (62.49 mg kg^−1^ DW), followed by the water (56.22 mg kg^−1^ DW) and S-nitroso-MSA (55.60 mg kg^−1^ DW) treatments, with MSA_NP providing the lowest levels (49.88 mg kg^−1^ DW) (Figure 10a). Moreover, under severe Cu stress, MSA_NP led to a 16% higher root Cu concentration than S-nitroso-MSA_NP (Figure 10b). 

In both stress conditions, the soil supplementation with Cu affected the concentration of the most essential elements in the roots (Figure 11). Under moderate Cu stress, the increase in Cu levels correlated with the increase in the Mg content and with the decrease in Fe, Ni, Co, and Mo contents (Figure 11a). Under severe Cu stress, the Cu accumulation in the roots correlated with the increase in S, Mn, and Co levels, and with the decrease in K, Ca, and B contents (Figure 11b).

Under moderate Cu stress, 64.2 and 29.1% of the variation were explained in the components 1 and 2, respectively (Figure 12a). The control correlated positively with Mo, Co, Fe, Ni, Zn, and Ca in the positive part of PC1, while the other treatments (plants submitted to cultivation in soil supplemented with Cu) were located in the negative part of PC1 (positive correlation with Cu, Mg, S, Mn, K, and P). Along PC2, MSA_NP (correlation mainly with P, K, Mn and S) separated from S-nitroso-MSA_NP, the water, and S-nitroso-MSA (correlation mainly with Cu and Mg). A cluster analysis by the hierarchical method showed the groups formed by dissimilarity (cophenetic correlation coefficient of 0.9606), ratifying the PCA (Figure 12b).

For severe Cu stress, 78.0 and 16.1% of the variation were explained in components 1 and 2, respectively (Figure 13a). Along PC1, there is a clear separation between the control (correlation with K, Ca, B, P, and Mo) and the other treatments (correlation with Cu, S, Co, Mg, and Mn), which separated along PC2. The cluster analysis showed the formation of groups (cophenetic correlation coefficient of 0.9782) similar to the other experiment under moderate Cu stress (Figure 13b).

## 3. Discussion

### 3.1. Characterization of the Nanoparticles and the Encapsulation Efficiency of MSA

The results of hydrodynamic size, PDI, and zeta potential of MSA_NP showed a particle in a homogenous size range with a positive zeta potential, due to the protonated amino groups from CS, which can increase the nanoparticle stability in biological environments. The hydrodynamic size of MSA_NP is in accordance with other similar studies from our research group [16,18]. Similarly, the TEM results indicated the spherical shape of the nanoparticles with a homogenous dispersion. Most nanoparticles observed by TEM were in the range of 10.4 to 12.8 nm at a solid state. Indeed, higher values of the hydrodynamic size of nanoparticles (measured by DLS) are expected, in comparison with the size at solid state (TEM measurements), due to the hydration of the nanoparticle surface [26]. The high value of the encapsulation efficiency of MSA into the nanoparticles indicates a successful formation of MSA_NP, which is attributed to the strong and positive electrostatic interactions of MSA with chitosan chains [27].

### 3.2. Morphophysiological Analyses

Applying similar growth conditions, soil type, and genotype to the present study, we demonstrated all of the effects triggered by different concentrations of Cu in the soil on soybean plants. In these previous studies, there was no impact of Cu stress on the total seed germination (%), but a delay in the germination process was observed in Cu concentrations greater than 117.89 mg kg^−1^ [10]. Furthermore, a dual effect of Cu on plant growth and development was verified, with lower concentrations of Cu providing better growth, compared to the control treatment (soil with a natural content of Cu), highlighting the concentration range from 90 to 98.42 mg kg^−1^ Cu. The decline began from 133.5 mg kg^−1^ Cu, with the incidence of oxidative stress starting at 164 mg kg^−1^ Cu (moderate) and with the highest values provided by the concentration of 244 mg kg^−1^ Cu (severe) [11]. In the present study, the similarity in the morphological parameters between the control and the water treatments under moderate Cu stress, indicates that the harmful effects triggered by the excess of Cu acts initially on the physiological and biochemical parameters of the plants. However, under severe Cu stress, the excess of Cu in the soil started to affect the morphological parameters, compromising plant growth and development (Table 1).

The S-nitroso-MSA_NP treatment promoted a more remarkable growth of soybean plants (length and dry mass) than the control and water treatments, under moderate Cu stress. In this condition, the treatment of S-nitroso-MSA and MSA_NP did not differ from the water treatment in most parameters evaluated (Table 1). Under severe Cu stress, the treatments could attenuate the harmful effects of the metal, with S-nitroso-MSA_NP promoting a better shoot and root development, when compared to the water treatment, while S-nitroso-MSA and MSA_NP induced only a better root development. Furthermore, under severe Cu stress, the formulations that release NO (S-nitroso-MSA_NP and S-nitroso-MSA), provided the same mass accumulation (RDW and SDW), compared to the control (Table 1). 

The application of NO donors is an efficient strategy to attenuate the effects of stress induced by different types of metals [12]. Varying the time of exposure to metals, plant species, and mechanism of action, most studies have showed a more significant biomass accumulation in plants treated with a NO donor and subjected to Cu stress, than in the control treatments [12]. Here, among the plants subjected to severe Cu stress, the improved development of the root system of the plants treated with S-nitroso-MSA_NP can be highlighted (Figure 2b), although free S-nitroso-MSA also exert some positive effects on plant growth (Table 1). In different situations of stress and cultivation, the development of the root system has been commonly reported as a result of the protection of NO donors against the stress induced by metals [12].

Regarding the PSII activity, it was possible to observe that the formulations prevented the Cu-induced decrease in F_v_/F_0_ (under severe Cu stress) and rETR (under moderate Cu stress) (Figure 3b,c). Moreover, the analysis of the leaf gas exchange highlighted the protective effect of the formulations that release NO. S-nitroso-MSA_NP and S-nitroso-MSA provided *A* similar to the control under Cu stress conditions (Figure 4a,b). However, under severe Cu stress, S-nitroso-MSA_NP stands out as the only treatment to maintain *g*_s_ at similar levels to the control (Figure 4d). S-nitroso-MSA_NP and S-nitroso-MSA were also efficient in maintaining the *k* at similar values to the control (Figure 5). Aftab et al. [28] observed higher *A*, *g*_s,_ and internal CO_2_ concentrations in *Artemisia annua* plants subjected to treatment with SNP (2 mM), aiming at the protection against aluminum stress. Treatment with SNP (100 µM) was adequate to increase the photosynthesis of different plant species, under cadmium-induced stress [29,30,31,32], nickel [33], and lead [34]. Dong et al. [35], evaluating the effects of treatment with SNP (100 µM) in plants of *Lolium perenne,* cultivated under Cu stress, observed the maintenance of the chlorophyll content and the photosynthetic activity, in addition to restoring the intracellular ionic balance. However, the increase in SNP concentrations (400 µM) did not reverse the inhibition caused by Cu and did not reduce the oxidative damage; on the contrary, it produced more toxic effects in plants.

In the present study, concentrations of 1 mM were used to compare the formulations (S-nitroso-MSA_NP; S-nitroso-MSA and MSA_NP) to the control and water treatments. However, the application via the soil may justify the protective effect of the treatments containing NO, even at relatively high concentrations, in comparison to the previous studies that used other treatment methods. However, the nanoencapsulation resulted in the improved effects of S-nitroso-MSA in protecting the morphophysiological parameters of soybean plants against Cu stress, which could be related to the gradual release of NO. Oliveira et al. [16] demonstrated that the CS nanoparticles containing S-nitroso-MSA were much more effective in mitigating the harmful effects of salt stress on the morphophysiological parameters of maize plants, when compared to the same NO donor in its free form, requiring a 50% lower concentration to ensure the same protective effect. In addition, the release rate of NO from the nanoencapsulated S-nitroso-MSA was on average 5.6 times slower than the release from the same donor in its free form.

### 3.3. Biochemical Analyses

In both Cu levels in the soil, there was a significant increase in the concentration of RSNO in the roots of plants subjected to treatment with S-nitroso-MSA_NP (Figure 6), evidencing the effectiveness of the treatment in increasing the endogenous NO bioavailability. Similar results were found by Lopes-Oliveira et al. [18] and do Carmo et al. [14]. RSNO may spontaneously release NO (thermal decomposition) and it can be triggered by light, Cu ions, and enzymes [36]. In the present study, the application of S-nitroso-MSA in its free form in a soil enriched with Cu, may have promoted a rapid decomposition of the molecule. Moreover, S-nitroso-MSA_NP may have benefited from the protection provided by the nanoencapsulation, preventing the rapid degradation of the molecule. The gradual release of NO may have favored the increase in the NO bioactivity in soybean plants. Moreover, the mucoadhesive property of CS may have favored the interaction of nanoparticles with the roots [22], improving the delivery of NO to the plant cells. As a consequence, the nanoencapsulation of S-nitroso-MSA provided a greater protection of the plants, with the decrease of both oxidative stress markers (H_2_O_2_ CD) in the roots of plants under severe Cu stress (Figure 7). In the same condition, S-nitroso-MSA_NP was the only formulation that reduced the POD activity (Figure 8d), which can be used as an additional biochemical marker of stress resulting from biotic and abiotic factors [37]. Sun et al. [38] found that the early NO burst provided by exogenous application of SNP (250 µM), plays a vital role in the Al resistance of *Triticum aestivum* cv. Yang-5/Jian-864 through modulation enhanced antioxidant defense, leading to the decreased H_2_O_2_ content and the lipid peroxidation of cells. Hu [39] showed that increasing the concentration of endogenous NO in the roots of *Hordeum vulgare* cv. Nude, provided by the treatment with SNP (200 µM), was the initial response through the imposition of stress by Cu, modulating the antioxidant defense and reducing the oxidative damage in the roots. 

In general, NO donors applied before or simultaneously to the metal treatment, regardless of metal type and plant species, attenuate the effects of the induced stress, increasing the antioxidant activity in the root and, consequently, promoting a more remarkable growth and reduction of the oxidative damage [12]. Here, comparing the formulations that release NO, S-nitroso-MSA_NP induced higher SOD and CAT activities than the S-nitroso-MSA treatment, which reinforces the improved protective effects of the nanoencapsulated NO donor against the oxidative stress induced by high Cu concentrations.

### 3.4. Nutrient Analysis

In both cultivation conditions, none of the treatments could avoid the excessive accumulation of Cu in the roots of the treated plants, compared to the control (Figure 10a,b). Accordingly, the multivariate analyses indicated a nutritional imbalance triggered by the excess of Cu in the soil (Figure 11, Figure 12 and Figure 13). Under moderate Cu stress, the elements Co, Fe, Ni, Mo, Zn, and Ca decreased in the root, due to the excess of Cu in the soil, evidenced by the highest concentrations in the control. Furthermore, the elements Mg, S, K, Mn, and P increased in the roots of plants subjected to stress. Among these elements, MSA_NP provided the highest P, S, K, and Mn concentrations and the lowest Cu levels. Moreover, S-nitroso-MSA_NP attenuated the impacts on the decrease of Zn and Ca (Figure 12). The increase of Cu in the root had a significant correlation with the rise of Mg and the reduction of Co, Mo, Ni, and Fe, impacting the interaction with the other elements (Figure 11a). In addition, the increase of some elements, such as S, in the treatments submitted to cultivation in soil supplemented with Cu may be associated with the accumulation of Cu in the roots (Figure 12a). The increase in S in the roots is related to the response of *Glycine max* cv. BRS 257 to Cu-induced stress, aiming at metal complexation [11]. Various metal chelators (such as phytochelatins and metallothioneins) have a high quantity of S in their composition [40,41]. In cultivation under moderate Cu stress, the highest concentrations of the elements in the roots of plants treated with MSA_NP may be related to the characteristics of chitosan. The application via soil may have favored the interaction of chitosan nanoparticles and MSA with different nutrients, changing the absorption of Cu and favoring the absorption of other elements due to synergism, antagonism, and inhibitions between the essential elements. According to Hidangmayum et al. [24], chitosan can form complexes with metals and is even used as a tool for soil bioremediation. Furthermore, Cu ions might have been chelated by the thiol group of MSA [42]. 

Under severe Cu stress, the root concentration of the K, Ca, and B decreased as a function of excess Cu in the soil, while the Mg, Mn, Co, and S content increased in the roots of plants subjected to stress (Figure 13). The rise in the Cu content in the root correlates significantly with the increase in Mn, Co, and S, and to the decrease in K, Ca, and B (Figure 11b). The essential macronutrients K and Ca are involved in several physiological and biochemical processes and play a fundamental role in mitigating different abiotic stresses [43]. The excess of Cu in the soil, decreasing the absorption of K and Ca and, consequently, the concentration of these elements in the roots, is probably one reason for the harmful effect on plants submitted to severe Cu stress, impacting the plant growth. Similar to moderate stress, the attenuation of the negative effect was observed in plants submitted to the other treatments (S-nitroso-MSA_NP; S-nitroso-MSA and MSA_NP) except the water treatment. Due to the chitosan properties and soil application, S-nitroso-MSA_NP and MSA_NP were able to attenuate the impacts of the excess of Cu in the absorption of K (Figure 13a). Furthermore, the increase of Mg in the roots of treated plants with S-nitroso-MSA_NP and S-nitroso-MSA (Figure 13b), might have contributed to an improved growth, development, and tolerance to excess Cu, provided by these treatments [44]. Similar to moderate Cu stress, the increase in Cu in the root is directly related to the increase in S, involved in the complexation of the metal. In both cultivation conditions, the water treatment showed the more severe nutritional imbalance caused by the absence of the effects conferred by the chitosan nanoparticles, NO and/or MSA.

In the present study, under moderate Cu stress, the excess of the metal affects most of the elements evaluated, generating direct impacts on the absorption of micronutrients. However, under severe Cu stress, the excess of the metal directly impacts the absorption of the macronutrients, significantly compromising the development and tolerance to excess Cu in the soil. As for NO, it is possible to find different reports in the literature about the fundamental role of NO in inducing responses in plants to stress by metals both by the endogenous concentration and by treatment with NO donors [12]. Moreover, the SNP as a NO donor, has commonly been used to investigate the effects of treatment with exogenous NO, demonstrating the potential to provide a more significant accumulation of biomass in treated plants, decreased metal absorption, a more significant antioxidant activity in the root, and control of the interaction between NO and ROS. Due to its low cost and ease of handling, SNP is the most used NO donor for plant applications. However, SNP is photosensitive, and its degradation is accompanied by the release of cyanide, leading to cellular toxicity [45]. Due to the nature of the molecules and the sensitivity to environmental factors [15], the treatment of plants with NO donors has limitations, mainly aimed at using the technique in less favorable environments, such as the agroecosystem. As an alternative, the nanoencapsulation of NO donors can make the exogenous application of NO an effective tool in protecting plants against different types of stress [14,16,18,19,20], enabling its use in agriculture.

The present study was the first to show the potential for using the nanoencapsulation of a NO donor as a strategy to protect plants against Cu stress. However, it is noteworthy that the results may vary, depending on the different factors, such as the diversity of soil types found in the producing regions. Due to the physical, chemical, and biological properties of the soil, the protective effect conferred by the soil treatment with these nanoparticles containing NO donors can be affected. Under the conditions used in this work, in almost all morphological parameters evaluated in the cultivation under moderate Cu stress, S-nitroso-MSA_NP promoted plant growth that surpassed even the control, with its plants cultivated under the natural conditions of Cu. With the increased Cu concentration in the soil, the S-nitroso-MSA_NP was again more effective than S-nitroso-MSA in ensuring plant growth. Among the treatments containing NO, only S-nitroso-MSA_NP guaranteed the maintenance of *g*_s_ in the condition of more significant stress, in addition to considerably increasing the bioavailability of NO in the roots, providing a more significant antioxidant activity, attenuation of the oxidative damage, and a greater capacity to mitigate the nutritional imbalance in the roots triggered Cu stress. From an economic point of view, the NO donor S-nitroso-MSA is a low-cost compound, compared to the NO donors usually applied to plants [15]. The association between NO donors and nanomaterials enables the exogenous application of these NO donors in agriculture, creating controlled release mechanisms capable of avoiding the toxic effects of high NO concentrations, as well as reducing the susceptibility to the environmental factors, consolidating itself as an innovative strategy to protect plants from different types of stress, including metal stress.

## 4. Materials and Methods

### 4.1. Synthesis of NO-Releasing CS Nanoparticles

NO-releasing CS nanoparticles were prepared following two steps. The first step consists of the synthesis of the CS nanoparticles containing mercaptosuccinic acid (MSA), the precursor of the NO donor. In the second step, there is the nitrosation of MSA, leading to the formation of the S-nitroso-MSA molecule, which is responsible for releasing NO.

CS nanoparticles containing 1 mM of MSA were prepared by the ionotropic gelation method. Briefly, a mixture of CS (26.7 mg mL^−1^) and MSA (0.4 mg mL^−1^) was prepared in acetic acid (1%, *v*/*v*) under magnetic stirring for 90 min at room temperature (pH ~ 4). A sodium tripolyphosphate (TPP) aqueous solution (2.4 mg mL^−1^) was dropped into the CS/MSA mixture using an automatic pump controller unit (Marte, MPV-500, São Paulo, SP, BR) at the flow rate of 132 µL min^−1^ using the volumetric proportion between CS/MSA and TPP of 3 to 1 [18].

The second step consists of the S-nitroso-MSA formation from the MSA encapsulated into chitosan nanoparticles or free MSA, for comparison. MSA was nitrosated by the reaction with an equimolar amount of sodium nitrite (NaNO_2_) for 1 h, protected from light. Free S-nitroso-MSA (S-nitroso-MSA) or S-nitroso-MSA encapsulated into the chitosan nanoparticles (S-nitroso-MSA_NP) were diluted to the desired concentrations and used immediately after the nitrosation process [18]. CS nanoparticles containing MSA molecules (MSA_NP), were used as an experimental control and also to characterize the nanoparticle size, morphology, and encapsulation efficiency.

### 4.2. Characterization of the Nanoparticles and the Determination of the Encapsulation Efficiency of MSA in the Nanoparticles 

The average hydrodynamic diameter (% by intensity), polydispersity index (PDI), and zeta potential of MSA_NP were characterized by dynamic light scattering (DLS) (Nano ZS Zetasizer, Malvern Instruments Co, Malvern, UK). Assessments were performed in triplicates of two independent experiments (n = 6) at 25 °C, using a fixed angle of 173° in disposable folded capillary zeta cells with a 10 mm path length in aqueous suspension [18]. The morphology of MSA_NP at a solid state was obtained by transmission electron microscopy (TEM) at 80 kV (Carl Zeiss 120 TEM, Zeiss International, Oberkochen, Germany) [46]. 

The encapsulation efficiency of MSA into CS nanoparticles was measured by quantifying the amount of MSA that was not encapsulated in the nanoparticle. A volume of 500 µL of MSA_NP was added to a Microcon centrifugal filter device (MWCO 10,000, Millipore, Darmstadt, Germany) and was centrifuged for 5 min at 1232× *g*. The eluted solution (non-encapsulated MSA) was incubated for 10 min with 2 mL of Ellman’s’ reagent (DTNB) solution (0.7 mM) in PBS buffer (pH 7.4) containing 1 mM of ethylenediaminetetraacetic acid (EDTA). The final mixture was settled into a quartz cuvette, and the intensity of the absorption band at 412 nm was measured in an UV−vis spectrophotometer (ε = 14,150 mol L^−1^ cm^−1^) (Agilent, model 8454, Palo Alto, CA, USA). The assessments were performed in triplicate and the percentage of the encapsulated MSA was determined [27].

### 4.3. Soil Contamination with Cu 

The experiments were carried out in a greenhouse belonging to the Department of Animal and Plant Biology of the State University of Londrina (UEL, Londrina, Brazil). An eutroferric red oxisol was used in the experiments. The previous characterization by a specialized laboratory (ITL, Londrina, Brazil) showed pH CaCl_2_ 5.0; available contents of Ca, Mg, K, P, and Al, respectively, 4.0, 1.8, 0.65, 0.024, and 0.04 cmol_c_ dm^−3^; organic matter 28.2 g kg^−1^; clay texture (6.85% of sand; 13.35% of silty, and 77.80% of clay), and natural Cu content of 11.2 mg kg^−1^. The soil contamination with two different Cu concentrations proceeded, according to the methodology explained by Gomes et al. [10,11]. For the determination of the bioavailable Cu, air-dried soil samples were used for extraction with Melich-1 solution [47], using an atomic absorption spectrophotometer, model 932 AA (GBC Scientific Equipment Ltd., Dandenong, Australia), equipped with a single element hollow cathode lamp, and air-acetylene burner (detection limit of 0.025 μg mL^−1^). The analytical calibration curve had a correlation coefficient *R*^2^ = 0.998, and all assessments were performed in quadruplicate. The bioavailable Cu contents detected in the contaminated soil were 164 and 244 mg Cu kg^−1^ of dry soil weight, which were considered the moderate and severe Cu stress conditions, respectively. Generally, the natural Cu content in soils is around 5 to 30 mg kg^−1^, but it can reach levels above 200 mg kg^−1^ in soils contaminated by excess metal [48]. 

### 4.4. Biological Material, Treatments and Sample Harvesting

Due to the increased use of Cu-based fungicides combined with systemic fungicides to optimize the Asian soybean rust management [49], soybean plants were selected as a model for this study. Seeds of a conventional soybean cultivar *(Glycine max* (L.) Merr. cv. BRS 257) were used in this study (kindly provided by the Brazilian Agricultural Research Corporation—Soybean, Londrina, Brazil). Two different experiments were carried out: soybean plants growing in a soil supplemented with 164 mg kg^−1^ Cu (moderate Cu stress) and 244 mg kg^−1^ Cu (severe Cu stress). In both experiments, four seeds were sown in plastic pots (1.16 L) filled with contaminated soil. As a control, there were also soybean plants treated with distilled water and maintained in soil without Cu addition. In each experiment, five experimental groups were used, as presented in Table 2. Following the sowing, the formulations (36 mL per pot) were applied weekly for three consecutive days, followed by four days without treatment. This frequency was used for three weeks, totaling nine days with and twelve days without treatment of the plants during the experiments. The plants were kept in a greenhouse without temperature and humidity control (with no shading). Tap water (100 mL) was applied when necessary to maintain the soil moisture close to field capacity. The experimental design used was completely randomized with eight replicates per treatment.

### 4.5. Morphophysiological Analyses 

One week after the end of the treatments (a total of 28 days after sowing), the photosynthetic assessments were performed in a randomly selected plant per pot when the plant had the first and second trefoil fully expanded. The photosynthetic variables were measured on the second trefoil.

Chlorophyll *a* fluorescence variables were measured in intact leaves using an OS1p fluorometer (Opti-Sciences, Hudson, NY, USA). The potential activity of photosystem II (PSII) was determined at dawn on dark-adapted leaves as the F_v_/F_0_ ratio, where F_v_ is the variable fluorescence and F_0_ is the basal fluorescence [50]. The effective quantum yield of PSII (ΔF/F_m_′) was measured at 10:30 am on light-adapted leaves, and the relative rate of linear electron transport of PSII (rETR) was calculated as rETR = ΔF/F_m_′ × PAR × 0.5 × 0.84 [50]. 

Gas exchange assessments were carried out from 9:00 am to 11:30 am using a portable infrared gas analyzer, LI-6400 XT model (LI-COR^©^ Biosciences, Lincoln, NE, USA), connected to a 6-cm^2^ chamber, adjusted for saturating the photosynthetically active radiation (PAR = 1500 μmol m^−2^ s^−1^). The net photosynthetic rate (*A*), stomatal conductance (*g*_s_), and instantaneous carboxylation efficiency (*k*) were determined [11]. 

Following the photosynthetic assessments, the four plants were removed from the pots, and the roots were washed with running water in a one-millimeter mesh sieve to completely remove the soil. Then, the roots were immersed for 5 min in a plastic container with 4 L of distilled water, and, still submerged, the roots of each plant were separated from each other. From the two plants randomly selected and destined for biochemical analyses, samples of the lateral roots without any soil trace were collected, immediately immersed in liquid nitrogen, and stored at −80 °C until the extraction to proceed with the analyses. The remaining two plants were used for the morphological analyses. The average value of these two plants was used for the measurements as the replicate. 

Shoot (SL) and root (RL) lengths were obtained with a measuring scale. For the dry weight determination, the plant material was separated into shoots (SDW) and roots (RDW) and dried in a drying oven at 60 °C until constant mass. The leaf area (LA) was obtained using a portable leaf area meter LI-3000C (LI-COR^©^ Biosciences, Lincoln, NE, USA).

### 4.6. Biochemical Analyses

The RSNO content in roots was determined as a NO bioavailability indicator. The total intracellular components were extracted through the homogenization with NEM (5 mM prepared in PBS, at pH 7.4) and sonication (45 kHz) for 10 min. The final homogenate was centrifuged (98,784× *g*) for 10 min and 20 μL of supernatant were added to 15 mL of the CuCl_2_ solution (100 mM) to enable the amperometric quantification of NO released, due to the RSNO decomposition [14,16,18,19]. Assessments were carried out in a free radical analyzer WPI TBR4100/1025 (World Precision Instruments Inc., Sarasota, FL, USA) equipped with a NO specific ISO-NOP sensor (2 mm). Data were compared to the standard curve obtained for GSNO. 

The occurrence of the oxidative stress was measured by determining the content of hydrogen peroxide (H_2_O_2_) and the conjugated dienes (CD), following the methodologies described by Alexieva et al. [51] and Boveris et al. [52], respectively. 

The activity of the antioxidant enzymes was determined using extracts obtained by homogenizing 0.1 g of the lateral root in 1 mL extraction buffer (1 mM EDTA, 0.1 M potassium phosphate buffer pH 7.5, 2% (*w*/*v*) polyvinylpolypyrrolidone), followed by centrifugation at 15,645× *g* (4 °C for 20 min). The superoxide dismutase activity (SOD, EC 1.15.1.1) was determined according to Giannopolitis and Reis [53], to measure the ability of the extract to inhibit the nitro blue tetrazolium chloride photoreduction. A unit of SOD was defined as the enzyme activity required to inhibit the nitro blue tetrazolium photoreduction by 50% (compared to the control). Peroxidase activity (POD, EC 1.11.1.7) was determined, according to Peixoto et al. [54] by following the increase in the absorbance at 420 nm, resulting from the pyrogallol oxidation in the presence of H_2_O_2_. The ascorbate peroxidase activity (APX, EC 1.11.1.11) was determined, according to Nakano and Asada [55] by monitoring the ascorbate consumption at 290 nm in the presence of H_2_O_2_. The catalase activity (CAT, EC 1.11.1.6) was determined according to Aebi et al. [56], Anderson et al. [57], and Peixoto et al. [54], by following the decrease in the H_2_O_2_ absorbance at 240 nm.

### 4.7. Nutrient Analysis

The same two plants selected for the morphophysiological analyses were used to provide material for the nutritional measurements. The quantification of Cu, Fe, Mn, Zn, Ni, Co, Mo, B, P, K, Ca, Mg, and S in the roots was performed using inductively coupled plasma mass spectrometry (ICP-MS), as explained in detail by Ondrasek et al. [58,59]. Briefly, ~0.1 g (dry weight) of the root was transferred to 15-mL tubes, supplemented with 1.5 mL of HNO_3_ (concentrated), and allowed to stand for approximately 48 h (pre-digestion stage). Then, the samples were heated in a water bath at 90 ± 5 °C for 4 h.

Finally, the tubes were filled with ultrapure water to the mark of 15 mL for the quantification of nutrients using a mass spectrometer (ICP-MS 7900, Agilent, Hachioji, Japan) equipped with a collision cell to minimize the isobaric and/or polyatomic interferences. High-purity argon (99.9999%, White Martins, Rio de Janeiro, Brazil) and helium (99.99%, White Martins, Rio de Janeiro, Brazil) were used for the plasma generation and as collision gas, respectively. The instrumental parameters used for the ICP-MS operation were based on previous studies [59,60]. 

To control the possible contamination of the reagents and the sample handling during the sample preparation procedure, four analytical blanks were prepared and submitted to the same steps as the samples. Yttrium was used as an internal standard to evaluate the equipment’s response during the analysis. The equipment calibration was performed by sequential dilution of a commercial multi-elemental standard solution containing 10 mg L^−1^ of the analytes in HNO_3_ 5% (*v*/*v*) (STD-3, PerkinElmer, Waltham, MA, USA). The range concentration of the calibration curves for all elements were from 1 to 200 μg L^−1^. The accuracy was assessed by analyzing the certified reference materials (CRMs: NIST 1573a (tomato leaves) from the National Institute of Standards and Technology (NIST); C1005a (sugarcane leaves) from the collaborative exercise CRM-Agro FC_012017, University of São Paulo (USP, São Paulo, BR) and the Brazilian Agricultural Research Corporation (EMBRAPA, Brasília, BR). The concentrations obtained for the elements analyzed in this study did not show significant differences (*t*-test, 90% confidence level) from the values reported in the certificates of these materials.

### 4.8. Statistical Analysis

Initially, the tests and graphical analyses of the residuals were carried out to verify the normality and homogeneity of variance. The results of the morphophysiological, biochemical, and Cu content determinations were submitted to ANOVA, followed by the Tukey test (*p* < 0.05). For the nutrient content in the roots, the average of the replicates was submitted to a Pearson correlation, principal component analysis (PCA), and a hierarchical clustering analysis [61]. Prior to carrying out the multivariate analyses, data normalization was performed to minimize the redundancy and inconsistencies. Based on PCA, a cluster analysis was carried out to verify which treatments were more similar to each other and to identify the subgroups of homogeneous regions concerning the nutritional content of the roots. The resulting hierarchical cluster was submitted to tests to obtain the cophenetic correlation coefficient and to validate the dissimilarity measure and chosen grouping method. All analyses were performed using the statistical program R [62], using the packages stats, easyanova, ExpDes.pt, biotools, RColorBrewer, FactoMineR, and factoextra.

## 5. Conclusions

The application of S-nitroso-MSA to the soil (in free and nanoencapsulated forms) was effective in promoting growth, protecting the root system, and enhancing the photosynthetic activity of *Glycine max* cv. BRS 257 under Cu stress. However, only the nanoencapsulated S-nitroso-MSA could significantly increase the bioavailability of NO in the roots, providing a more effective antioxidant protection and mitigation of the deleterious effects induced by Cu excess on soybean plants than its free form.

## Figures and Tables

**Figure 1 plants-11-03245-f001:**
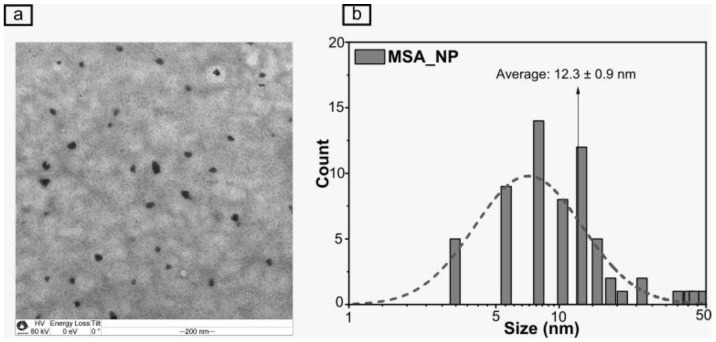
Representative transmission electron microscopy image of the chitosan nanoparticles loaded with mercaptosuccinic acid (MSA_NP) (**a**) and the corresponding size histogram of MSA_NP (**b**).

**Figure 2 plants-11-03245-f002:**
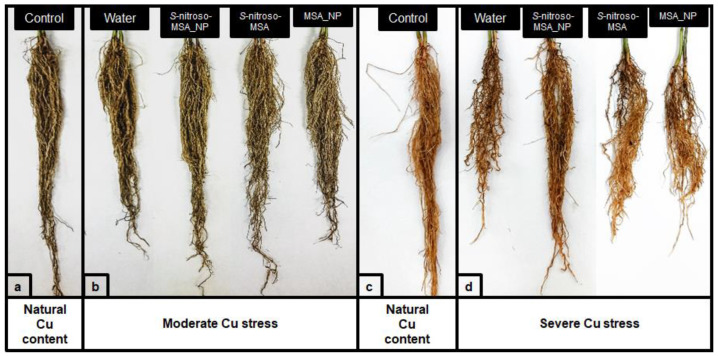
Visual aspect of the roots from soybean plants cultivated in soil with natural copper (Cu) content (**a**) and Cu supplementation to 164 mg kg^−1^ (**b**) in experiment 1. Visual aspect of the roots from the soybean plants cultivated in soil with a natural Cu content (**c**) and Cu supplementation of 244 mg kg^−1^ (**d**) in experiment 2. Control = treatment with distilled water; water = treatment with distilled water after the Cu addition; S-nitroso-MSA_NP = treatment with a nanoencapsulated NO donor (S-nitroso-mercaptosuccinic acid) at 1 mM; S-nitroso-MSA = treatment with a NO donor (S-nitroso-mercaptosuccinic acid) in the free form at 1 mM; MSA_NP = treatment with nanoparticles containing the non-nitrosated MSA at 1 mM.

**Figure 3 plants-11-03245-f003:**
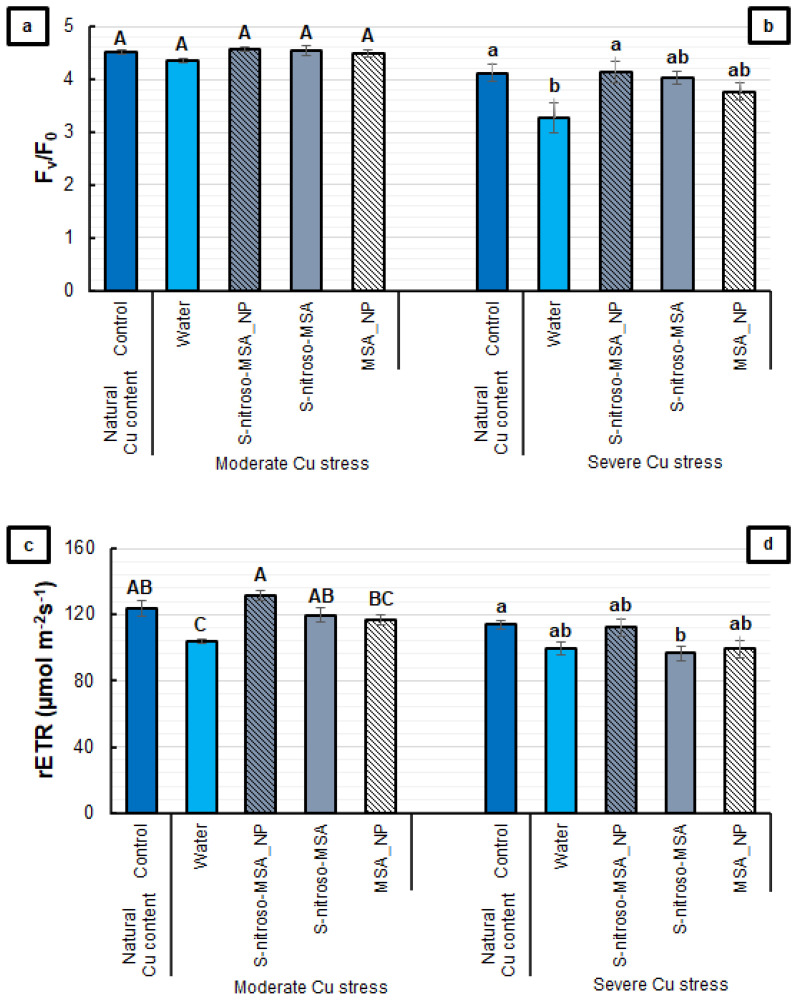
Potential activity of photosystem II (F_v_/F_0_) (**a**,**b**) and the relative rate of the linear electron transport of photosystem II (rETR) (**c**,**d**) of soybean plants cultivated in soil containing different copper (Cu) levels (natural content: 11 mg kg^−1^; moderate Cu stress: 164 mg kg^−1^; severe Cu stress: 244 mg kg^−1^). Control = treatment with distilled water; water = treatment with distilled water after the Cu addition; S-nitroso-MSA_NP = treatment with a nanoencapsulated NO donor (S-nitroso-mercaptosuccinic acid) at 1 mM; S-nitroso-MSA = treatment with a NO donor (S-nitroso-mercaptosuccinic acid) in the free form at 1 mM; MSA_NP = treatment with nanoparticles containing the non-nitrosated MSA at 1 mM. Data are the mean ± standard error (*n* = 8). The same uppercase and lowercase letters on the column (referring to experiments 1 and 2, respectively) indicate values that do not differ by ANOVA, followed by the Tukey test (*p* < 0.05).

**Figure 4 plants-11-03245-f004:**
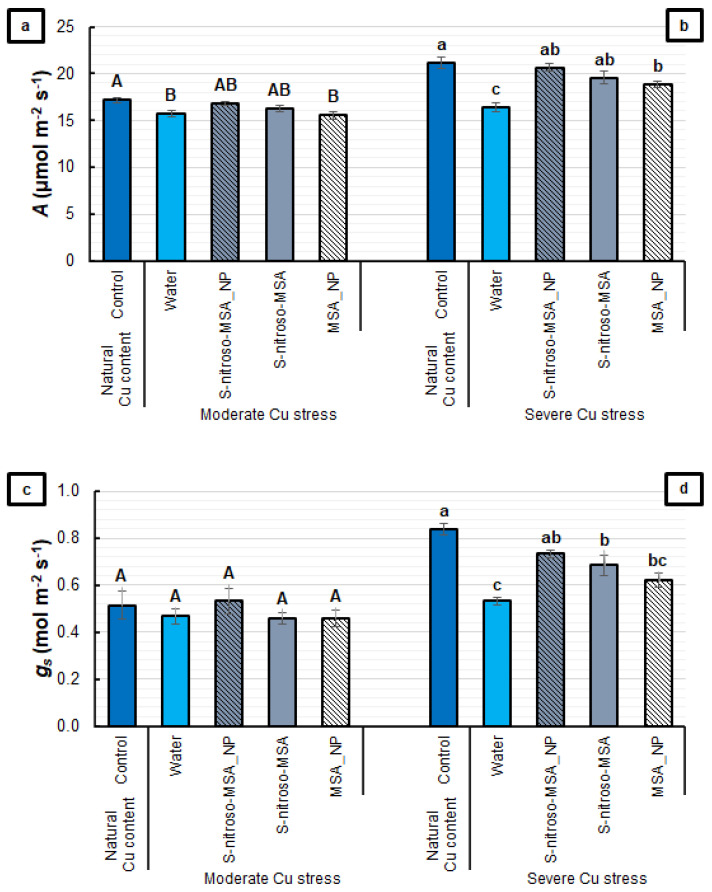
Net photosynthetic rate (*A*) (**a**,**b**) and stomatal conductance (*g*_s_) (**c**,**d**) of soybean plants cultivated in soil containing different copper (Cu) levels (natural content: 11 mg kg^−1^; moderate Cu stress: 164 mg kg^−1^; severe Cu stress: 244 mg kg^−1^). Control = treatment with distilled water; water = treatment with distilled water after the Cu addition; S-nitroso-MSA_NP = treatment with a nanoencapsulated NO donor (S-nitroso-mercaptosuccinic acid) at 1 mM; S-nitroso-MSA = treatment with a NO donor (S-nitroso-mercaptosuccinic acid) in the free form at 1 mM; MSA_NP = treatment with nanoparticles containing the non-nitrosated MSA at 1 mM. Data are the mean ± standard error (*n* = 8). The same uppercase and lowercase letters on the column (referring to experiments 1 and 2, respectively) indicate values that do not differ by ANOVA, followed by the Tukey test (*p* < 0.05).

**Figure 5 plants-11-03245-f005:**
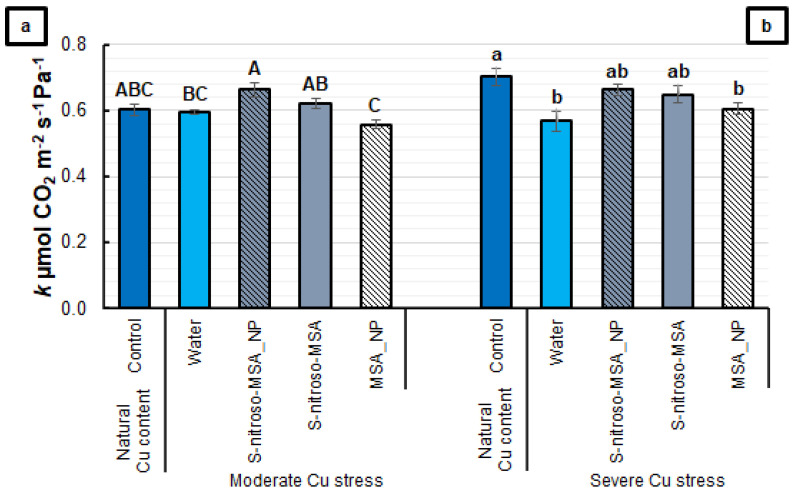
Instantaneous carboxylation efficiency (*k*) (**a**,**b**) of soybean plants cultivated in soil containing different copper (Cu) levels (natural content: 11 mg kg^−1^; moderate Cu stress: 164 mg kg^−1^; severe Cu stress: 244 mg kg^−1^). Control = treatment with distilled water; water = treatment with distilled water after Cu addition; S-nitroso-MSA_NP = treatment with a nanoencapsulated NO donor (S-nitroso-mercaptosuccinic acid) at 1 mM; S-nitroso-MSA = treatment with a NO donor (S-nitroso-mercaptosuccinic acid) in the free form at 1 mM; MSA_NP = treatment with nanoparticles containing the non-nitrosated MSA at 1 mM. Data are the mean ± standard error (*n* = 8). The same uppercase and lowercase letters on the column (referring to experiments 1 and 2, respectively) indicate values that do not differ by ANOVA, followed by the Tukey test (*p* < 0.05).

**Figure 6 plants-11-03245-f006:**
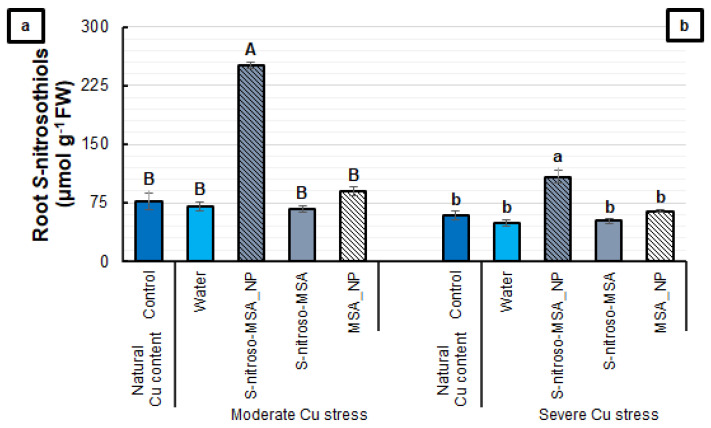
Root S-nitrosothiol content (**a**,**b**) of soybean plants cultivated in soil containing different copper (Cu) levels (natural content: 11 mg kg^−1^; moderate Cu stress: 164 mg kg^−1^; severe Cu stress: 244 mg kg^−1^). Control = treatment with distilled water; water = treatment with distilled water after Cu addition; S-nitroso-MSA_NP = treatment with a nanoencapsulated NO donor (S-nitroso-mercaptosuccinic acid) at 1 mM; S-nitroso-MSA = treatment with a NO donor (S-nitroso-mercaptosuccinic acid) in the free form at 1 mM; MSA_NP = treatment with nanoparticles containing the non-nitrosated MSA at 1 mM. Data are the mean ± standard error (*n* = 4). The same uppercase and lowercase letters on the column (referring to experiments 1 and 2, respectively) indicate values that do not differ by ANOVA, followed by the Tukey test (*p* < 0.05).

**Figure 7 plants-11-03245-f007:**
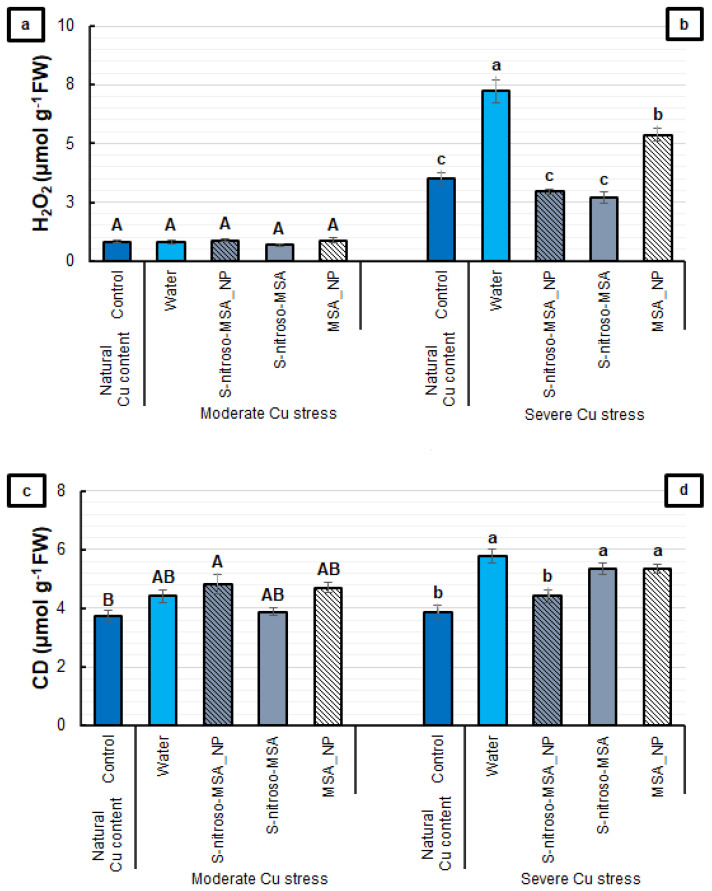
Root H_2_O_2_ (**a**,**b**) and conjugated dienes (CD) (**c**,**d**) contents of soybean plants cultivated in soil containing different copper (Cu) levels (natural content: 11 mg kg^−1^; moderate Cu stress: 164 mg kg^−1^; severe Cu stress: 244 mg kg^−1^). Control = treatment with distilled water; water = treatment with distilled water after Cu addition; S-nitroso-MSA_NP = treatment with a nanoencapsulated NO donor (*S*-nitroso-mercaptosuccinic acid) at 1 mM; S-nitroso-MSA = treatment with a NO donor (S-nitroso-mercaptosuccinic acid) in the free form at 1 mM; MSA_NP = treatment with nanoparticles containing the non-nitrosated MSA at 1 mM. Data are the mean ± standard error (*n* = 4). The same uppercase and lowercase letters on the column (referring to experiments 1 and 2, respectively) indicate values that do not differ by ANOVA, followed by the Tukey test (*p* < 0.05).

**Figure 8 plants-11-03245-f008:**
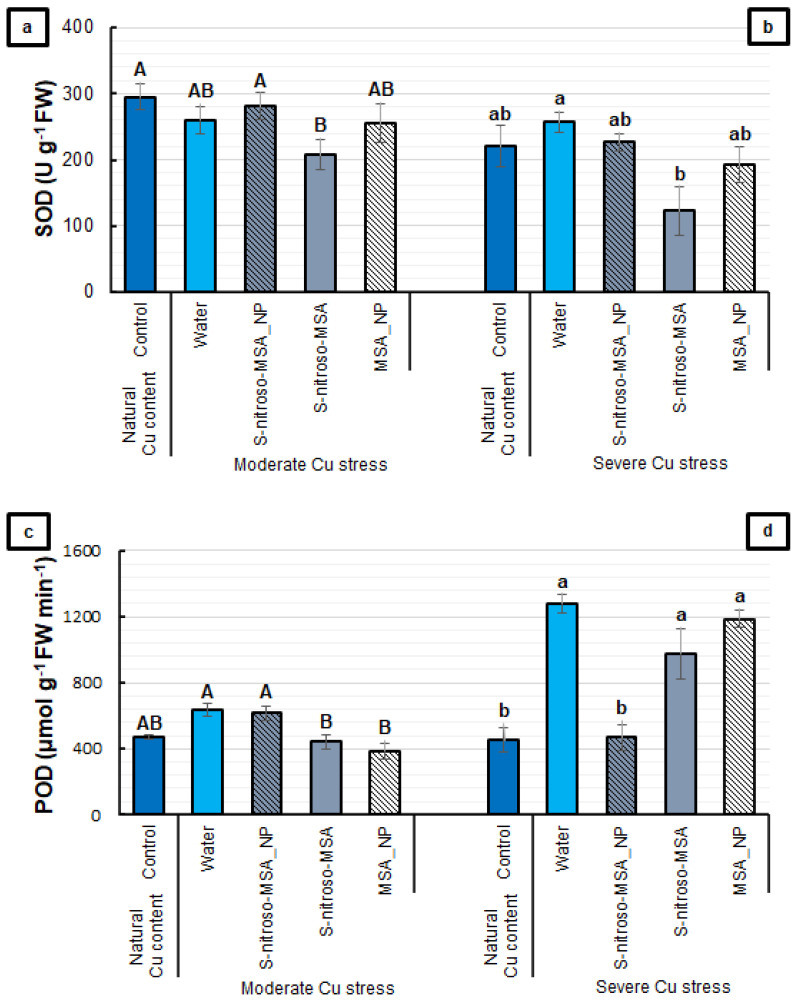
Root superoxide dismutase (SOD) (**a**,**b**) and peroxidase (POD) (**c**,**d**) activities of soybean plants cultivated in soil containing different copper (Cu) levels (natural content: 11 mg kg^−1^; moderate Cu stress: 164 mg kg^−1^; severe Cu stress: 244 mg kg^−1^). Control = treatment with distilled water; water = treatment with distilled water after Cu addition; S-nitroso-MSA_NP = treatment with a nanoencapsulated NO donor (*S*-nitroso-mercaptosuccinic acid) at 1 mM; S-nitroso-MSA = treatment with a NO donor (S-nitroso-mercaptosuccinic acid) in the free form at 1 mM; MSA_NP = treatment with nanoparticles containing the non-nitrosated MSA at 1 mM. Data are the mean ± standard error (*n* = 4). The same uppercase and lowercase letters on the column (referring to experiments 1 and 2, respectively) indicate values that do not differ by ANOVA, followed by the Tukey test (*p* < 0.05).

**Figure 9 plants-11-03245-f009:**
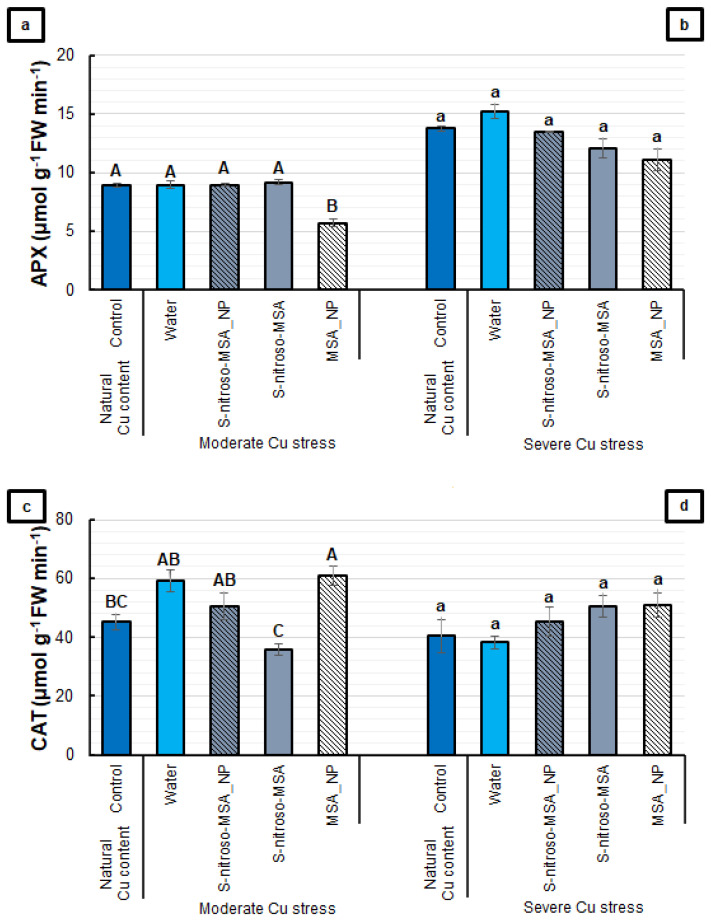
Root ascorbate peroxidase (APX) (**a**,**b**) and catalase (CAT) (**c**,**d**) activities of soybean plants cultivated in soil containing different copper (Cu) levels (natural content: 11 mg kg^−1^; moderate Cu stress: 164 mg kg^−1^; severe Cu stress: 244 mg kg^−1^). Control = treatment with distilled water; water = treatment with distilled water after Cu addition; S-nitroso-MSA_NP = treatment with a nanoencapsulated NO donor (S-nitroso-mercaptosuccinic acid) at 1 mM; S-nitroso-MSA = treatment with a NO donor (S-nitroso-mercaptosuccinic acid) in the free form at 1 mM; MSA_NP = treatment with nanoparticles containing the non-nitrosated MSA at 1 mM. Data are the mean ± standard error (*n* = 4). The same uppercase and lowercase letters on the column (referring to experiments 1 and 2, respectively) indicate values that do not differ by ANOVA, followed by the Tukey test (*p* < 0.05).

**Figure 10 plants-11-03245-f010:**
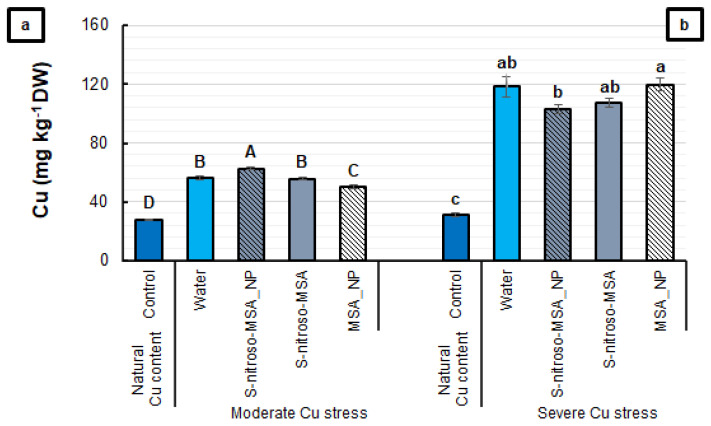
Root Cu content (**a**,**b**) of soybean plants cultivated in soil containing different copper (Cu) levels (natural content: 11 mg kg^−1^; moderate Cu stress: 164 mg kg^−1^; severe Cu stress: 244 mg kg^−1^). Control = treatment with distilled water; water = treatment with distilled water after Cu addition; S-nitroso-MSA_NP = treatment with a nanoencapsulated NO donor (S-nitroso-mercaptosuccinic acid) at 1 mM; S-nitroso-MSA = treatment with a NO donor (S-nitroso-mercaptosuccinic acid) in the free form at 1 mM; MSA_NP = treatment with nanoparticles containing the non-nitrosated MSA at 1 mM. Data are the mean ± standard error (*n* = 4). The same uppercase and lowercase letters on the column (referring to experiments 1 and 2, respectively) indicate values that do not differ by ANOVA, followed by the Tukey test (*p* < 0.05).

**Figure 11 plants-11-03245-f011:**
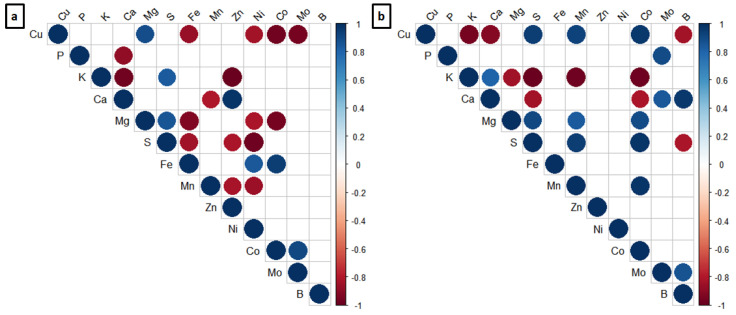
Corrplot depicting the correlation coefficient of the essential nutrients in the roots of soybean plants treated and cultivated in soil with natural Cu content (Control) and 164 mg kg^₋1^ (**a**) and natural Cu content (Control) and 244 mg kg^₋1^ (**b**). The areas of the circles show the value of the corresponding Pearson correlation coefficients. The significant (*p* < 0.05; *p* < 0.10) correlation values are presented in the upper panel in circle; The non-significant (*p* > 0.10) correlation values are present in the upper panel with a blank square; the positive correlations are displayed in blue and the negative correlations in red. Data are the average of the four replicates per treatment.

**Figure 12 plants-11-03245-f012:**
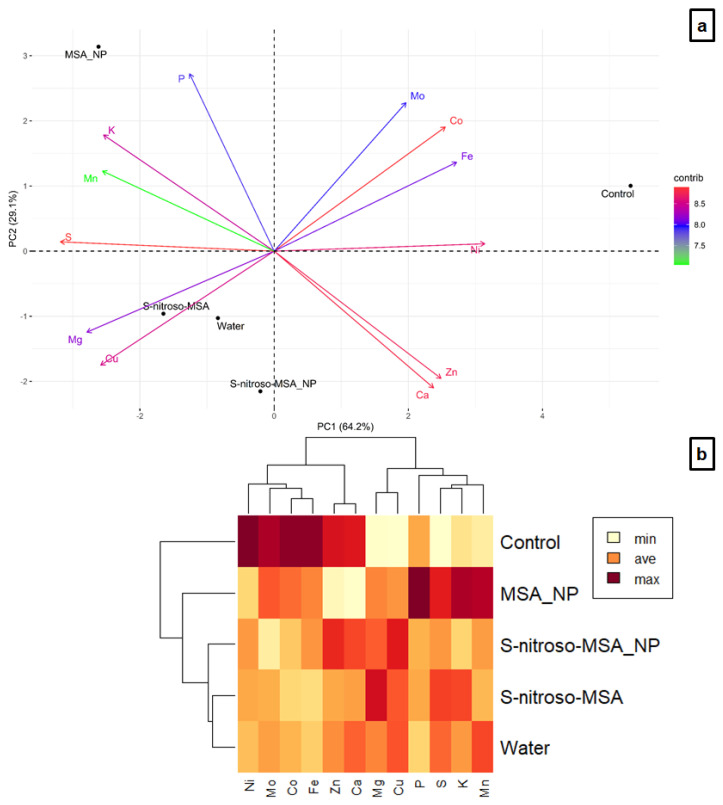
Principal component analysis (**a**), and heatmap hierarchical clustering analysis (**b**) of the essential nutrients in the roots of soybean plants cultivated in soil, containing different copper (Cu) levels (natural content: 11 mg kg^₋1^; moderate Cu stress: 164 mg kg^₋1^). Control = treatment with distilled water and soil with natural copper content; water = treatment with distilled water after Cu addition; S-nitroso-MSA_NP = treatment with a nanoencapsulated NO donor (S-nitroso-mercaptosuccinic acid) at 1 mM and soil with copper supplementation; S-nitroso-MSA = treatment with a NO donor (S-nitroso-mercaptosuccinic acid) in the free form at 1 mM and soil with copper supplementation; MSA_NP = treatment with nanoparticles containing the non-nitrosated MSA at 1 mM and soil with copper supplementation. Data are the average of the four replicates per treatment.

**Figure 13 plants-11-03245-f013:**
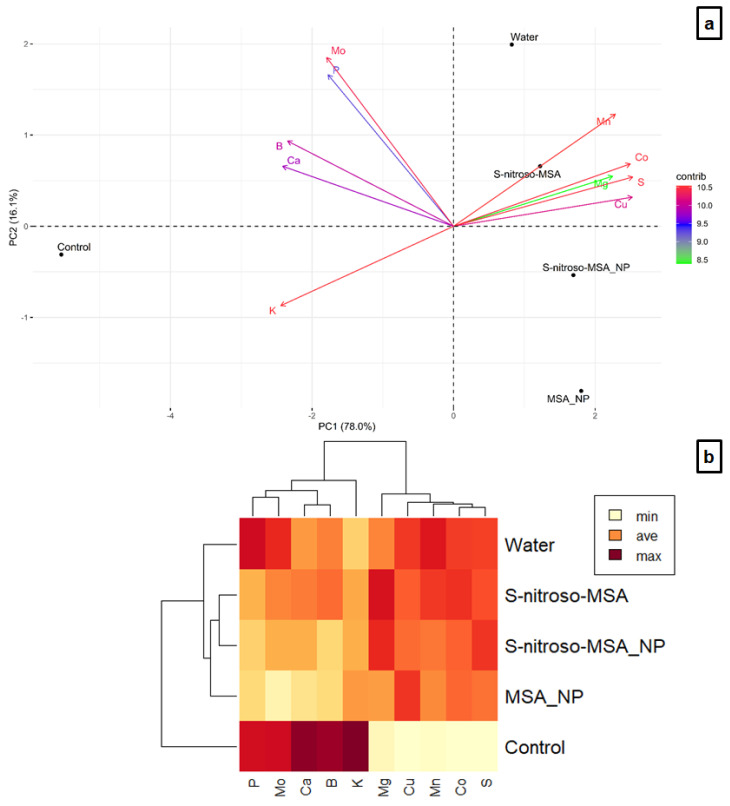
Principal component analysis (**a**), and heatmap hierarchical clustering analysis (**b**) of the essential nutrients in the roots of soybean plants cultivated in soil containing different copper (Cu) levels (natural content: 11 mg kg^₋1^; severe Cu stress: 244 mg kg^₋1^). Control = treatment with distilled water and soil with natural copper content; water = treatment with distilled water after Cu addition; S-nitroso-MSA_NP = treatment with a nanoencapsulated NO donor (S-nitroso-mercaptosuccinic acid) at 1 mM and soil with copper supplementation; S-nitroso-MSA = treatment with a NO donor (S-nitroso-mercaptosuccinic acid) in the free form at 1 mM and soil with copper supplementation; MSA_NP = treatment with nanoparticles containing the non-nitrosated MSA at 1 mM and soil with copper supplementation. Data are the average of the four replicates per treatment.

**Table 1 plants-11-03245-t001:** Root length (RL), root dry weight (RDW), shoot length (SL), shoot dry weight (SDW), and leaf area (LA) of the soybean plants grown in soil with a natural copper (Cu) content and in soil supplemented with Cu.

	Natural Cu Content		Cu Supplementation (164 mg kg^−1^)							
**Variables**	**Control ^1^**		**Water**		**S-nitroso-MSA_NP**		**S-nitroso-MSA**		**MSA_NP**	
RL (cm)	29.21 ± 0.83 ^2^	b ^3^	28.84 ± 0.71	b	33.14 ± 0.97	a	31.82 ± 0.75	ab	28.63 ± 0.67	b
RDW (g)	0.2073 ± 0.01	c	0.2101 ± 0.01	bc	0.2461 ± 0.01	a	0.2314 ± 0.01	ab	0.2440 ± 0.01	a
SL (cm)	17.51 ± 0.28	b	16.97 ± 0.31	b	19.02 ± 0.23	a	17.26 ± 0.42	b	17.36 ± 0.31	b
SDW (g)	0.4351 ± 0.01	b	0.4312 ± 0.01	b	0.4798 ± 0.01	a	0.4506 ± 0.01	ab	0.4502 ± 0.01	ab
LA (cm^2^)	143.27 ± 3.02	ab	137.13 ± 2.04	ab	146.72 ± 2.91	a	143.56 ± 3.78	ab	134.32 ± 3.32	b
	**Natural Cu content**		**Cu supplementation (244 mg kg^−1^)**							
**Variables**	**Control**		**Water**		**S-nitroso-MSA_NP**		**S-nitroso-MSA**		**MSA_NP**	
RL (cm)	28.72 ± 0.86	a	15.32 ± 0.88	c	20.63 ± 0.67	b	20.59 ± 1.03	b	20.36 ± 0.45	b
RDW (g)	0.1578 ± 0.01	a	0.1134 ± 0.01	c	0.1488 ± 0.01	ab	0.1418 ± 0.01	ab	0.1358 ± 0.01	b
SL (cm)	17.33 ± 0.37	a	14.47 ± 0.41	c	16.62 ± 0.18	ab	15.64 ± 0.35	bc	15.01 ± 0.18	c
SDW (g)	0.3658 ± 0.01	a	0.3286 ± 0.01	b	0.3624 ± 0.01	a	0.3448 ± 0.01	ab	0.3317 ± 0.01	b
LA (cm^2^)	116.74 ± 4.90	a	95.92 ± 7.51	b	110.56 ± 2.41	ab	102.13 ± 3.80	ab	94.54 ± 4.06	b

^1^ Control = treatment with distilled water; water = treatment with distilled water after the Cu addition; S-nitroso-MSA_NP = treatment with nanoencapsulated NO donor (S-nitroso-mercaptosuccinic acid) at 1 mM; S-nitroso-MSA = treatment with NO donor (S-nitroso-mercaptosuccinic acid) in the free form at 1 mM; MSA_NP = treatment with nanoparticles containing the non-nitrosated MSA at 1 mM. ^2^ Data are the mean ± standard error (*n* = 8). ^3^ Means followed by the same letter on the line do not differ by ANOVA, followed by the Tukey test (*p* < 0.05).

**Table 2 plants-11-03245-t002:** Treatments applied to the soybean plants.

Experimental Group	Soil ^1^	Formulation
Control	Natural Cu content	Distilled water
Water	Cu supplementation	Distilled water
S-nitroso-MSA_NP	Cu supplementation	Chitosan/sodium tripolyphosphate nanoparticles containing S-nitroso-MSA (1 mM)
S-nitroso-MSA	Cu supplementation	S-nitroso-MSA in the free form (1 mM)
MSA_NP	Cu supplementation	Nanoparticles containing the non-nitrosated MSA (1 mM)

^1^ The soil was supplemented with 164 mg Cu kg^−1^ in experiment 1 (moderate Cu stress) and 244 mg Cu kg^−1^ in experiment 2 (severe Cu stress).

## Data Availability

The data supporting reported results can be found through a direct request to the corresponding authors.

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
