# Peer review of "Soil Treatment with Nitric Oxide-Releasing Chitosan Nanoparticles Protects the Root System and Promotes the Growth of Soybean Plants under Copper Stress"

_plants, 2022, doi:10.3390/plants11233245_

Round 1
Reviewer 1 Report
The submitted manuscript presents the results of two experiments evaluating the efficiency of free and nanoencapsulated nitric oxide donors in alleviating the Cu induced stress in soybean cultivation
I find the manuscript well prepared, the design of the experiment was set up correctly. The study inlcuded wide range of performed asessments and analyses, the statistical verification of obtained results was performed correctly. I have no major concerns regarding the scientific merity of the proposed manuscript. Some detailed, more editorial comments are presented in the attached file.
I would also suggest careful rereading of the whole text in order to correct some grammatical errors that occurred.
Overall, I can recommend the manuscript to be published after minor corrections as proposed.
Reviewer 2 Report
att

Reviewer 3 Report
In the present paper, Gomes et al reported the synthesis and the application of novel NO-releasing chitosan-based nanoparticles (NP) potentially capable to contrast the Cu stress induced in plants. In detail, the authors evaluated the action of these NP on Glycine max (L.) soybeans grown in acidic and clay soil containing different doses of copper, under greenhouse condition.
Briefly, the authors conducted a NP characterization through TEM and DLS techniques. Then, evaluated the action of NPs by measuring in treated soybean plants (i) morphophysiological parameters (Fv/F0 ratio, rETR, photosynthetic rate, stomatal conductance), (ii) the most important biomarkers indicating the occurrence of a stress response, (iii) root nutrients content.
The paper is very interesting, is based on a relatively innovative idea and is well written. The experimental design is very appropriate as well as the results are promising. The topic is very suitable not only for the special issue, but also for a typical Plants reader.
However, I recommend a MAJOR REVISION accounting for the points listed as follows.
General comments:
- A limitation of the work is that the authors based their general conclusions only on data resulting from the application of the experimental design on a single soil type (moreover characterized by not common properties, such as very high clay content and very acid soil). I expect that soil properties, such as pH and texture, can play - somehow - a role in the action of proposed NP (i.e. acid soils may develop positive charges determining a repulsion toward the NPs characterized by a positive zeta potential; clay soils may retard the NP leaching as well as the chitosan decomposition of aerobic microbiota, and so on…). Therefore, the authors should, at least, include few comments on these aspects, by introducing the possibility that a different response could be expected by repeating the experiment in different soil types (i.e. sandy and subalkaline soils).
- NP characterization: Actually, the Figure 1a does not allow one to appreciate so many morphological details as to confirm the inference on NPs shape (page13, line 365…). I suggest to replace it with a figure characterized by a larger zoom of NPs. Moreover, the authors should explain how they measured an hydrodynamic radius of MSA_NP corresponding to 128.5 nm (page 2, line 91)… Presumably it is an error as the figure 1 b indicates an average size of 12.3 nm. Furthermore, a solid or semi-solid state molecular characterization (CPMAS NMR, HRMAS NMR, IR and so on…) of both synthesized MSA_NP and S-Nitroso MSA_NP would further strengthen the results.
- The authors evaluated the nutrients content in roots of treated plants. Although they conducted a very complete set of analysis, the relative discussion part is relatively poor and redundant and just is limited to observe a sort of different equilibrium vaguely ascribable to the Cu content response … I invite to improve the part. Moreover, although heatmap and PCA are very appealing and eye-catching statistical exercises (Figures 12 and 13) it seems that either the authors acquired and employed for these tests only a single measure per nutrient and per treatment or calculated the average of the replicates (???...the authors did not provide sufficient information in Materials and method section). In all cases, it is meaningless to apply PCA and/or HEATMAP on a single observation per treatment… the corrplot in Figure 11 is more than enough. Then, even though the authors highlighted the fact that, as shown in Figures 12 and 13, the response of water treatment resulted isolated as compared to that induced by other treatments, they should also provide a comment on the fact that also MSA-NP induced a singular response in both cases.
- The authors used a 1mM concentration of NP treatments. How did the authors choose such a concentration? May the concentration be related to the NPs efficiency? The author should include a sentence on this aspect.
- The authors must provide more information on corrplot, HEATMAP and PCA in Materials and method section (I invite to remove part of details from the very long captions of Figures 11, 12 and 13)
Minor revisions:
Table 1: the authors should remove the bold case in the upper left side of the table as well as they should presumably add LOW Cu supplementation (up) and HIGH Cu supplementation (down).
Figure 2: The authors should have added a millimetric reference to the figure to make more reliable the comparison among all treatments…
Page 4, line 122. …whereas the roots..
In most of figure captions, I suggest to unequivocally distinguish “Control” from ”Water” description by modifying the second as follows: “treatment with distilled water after Cu addition”.
Page 8, line 220. Replace “uunder” with “under”
Page 16, line 390 separate …plants (length…)
Page 16, lines 374-377 The germination data are missing. They should include them.
Round 2
Reviewer 3 Report
The authors integrated and modified the manuscript by taking into account all of my indications/suggestions as well as their replies to my concerns were convincing and satisfying.
The manuscript may be accepted in the present form.